# Replication Study of "Fairness and Bias in Online Selection"

## Reproducibility Summary

**Scope of Reproducibility**

This report aims to reproduce the results in the paper *Fairness and Bias in Online Selection* [9]. The paper presents optimal and fair alternatives for existing Secretary and Prophet algorithms. Reproducing the paper involves validating three claims made by the authors [9]: (1) The presented baselines are either unfair or have low performance, (2) The proposed algorithms are perfectly fair, and (3) The proposed algorithms perform comparably to or even better than the presented baselines.

**Methodology**

We recreate the algorithms and perform experiments to validate the authors' initial claims for both problems under various settings, with the use of both real and synthetic data. The authors conducted the experiments in the C++ programming language. We largely used the paper as a resource to reimplement all algorithms and experiments from scratch in Python, only consulting the authors' code base when needed.

**Results**

For the Multi-Color Secretary problem, we were able to recreate the outcomes, as well as the performance of the proposed algorithm (with a margin of 3-4%). However, one baseline within the second experiment returned different results, due to inconsistencies in the original implementation. In the context of the Multi-Color Prophet problem, we were not able to exactly reproduce the original results, as the authors ran their experiments with twice as many runs as reported. After correcting this, the original outcomes are reproduced.

A drawback of the proposed prophet algorithms is that they only select a candidate in 50-70% of cases. None-result are often undesirable, so we extend the paper by proposing adjusted algorithms that pick a candidate (almost) every time. Furthermore, we show empirically that these algorithms maintain similar levels of fairness.

**What was easy**

The paper provides pseudocode for the proposed algorithms, making the implementation straightforward. More than that, recreating their synthetic data experiments was easy due to providing clear instructions.

**What was difficult**

However, we did run into several difficulties: 1) There were a number of inconsistencies between the paper and the code, 2) Several parts of the implementation were missing in the code base, and 3) The secretary experiments required running the algorithm over one billion iterations which makes verifying its results within timely manner difficult.

**Communication with original authors**

The authors of the original paper were swift in their response with regard to our findings. Our main allegations regarding inconsistencies in both the Secretary and Prophet problems were confirmed by the authors.

---

## 1 Introduction

Online selection is challenging, as candidates arrive sequentially and decisions need to be made with incomplete information. Such problems affect our lives in a profound way. Algorithmic credit scores determine who receives a mortgage or who can start a business. Automatic systems even decide who receives an organ transplant or who has priority in being admitted to an Intensive Care Unit. The growing importance of algorithms has prompted concerns about fairness alongside efficiency. Addressing these issues is essential. The authors focus on online selection problems where suited candidates have to be chosen with imperfect information.

In the paper "Fairness and Bias in Online selection" [9] three fair selection algorithms are presented to be applied to the domain of online selection. In particular, they make use of two well-known problems from the literature: the Secretary and the Prophet problems. Research has long focused on the performance of such algorithms, but fairness has received increasing attention recently.[1, 2] The authors of this paper aim to fill this gap. They consider an algorithm fair when "the solution obtained is balanced with respect to some sensitive attribute (e.g. nationality, race, gender)" [9]. This notion is consistent with previous work [3, 4, 5, 6, 7, 13]. More precisely, algorithms are considered fair when they respect the prior probability $p$ that the best candidate belongs to a certain group.

The aim of this study is to validate the claims made in the paper and see if their results can be reproduced. Furthermore, we propose an adjusted version of their prophet algorithms, in order to reduce the number of occurrences where the algorithm does not pick any candidate.

## 2 Scope of reproducibility

The focus of our study is to empirically verify the claims of the paper. The mathematical proofs and theorems lie outside our scope. The authors of the original paper propose three online algorithms, namely one Multi-Color Secretary and two Multi-Color Prophet approaches that are both fair and efficient. They also present several existing baseline algorithms. The main empirical claims of the paper are:

a) The used baselines are either unfair or have low performance.

b) The proposed algorithms are perfectly fair.

c) The proposed algorithms perform comparably or even better than the presented baselines.

These claims are supported by experiments where all algorithms select a candidate based on a variety of datasets. Performance is defined as the probability of selecting the optimal candidate (in the Secretary setting) and the average value of the selected candidate (in the Prophet setting). We validate their claims by recreating the algorithms and experiments from scratch in Python. We use the description in the paper as a guideline, referring to the original code only when necessary.

## 3 Methodology

### 3.1 The Secretary Algorithm

**Algorithm 1** describes the original authors' fair secretary algorithm. Candidates appear in increasing order of their arrival time $\tau$. The algorithm calculates one threshold $t \in [0, 1]^k$ per group $c$, using Formula 1.

$$t_k^* = (1 - (k-1)p_k)^{\frac{1}{k-1}}$$

$$t_j^* = t_{j+1}^* \left( \frac{\sum_{r=1}^{j} \frac{p_r}{j-1} - p_j}{\sum_{r=1}^{j} \frac{p_r}{j-1} - p_{j+1}} \right)^{\frac{1}{j-1}}, for \ 2 \leq j \leq k-1$$

$$t_1^* = t_2^* \cdot e^{\frac{p_2}{p_1} - 1}$$

(1)

---

**Algorithm 1** Fair Secretary

**Input:** $\mathbf{t} \in [0, 1]^k$, a time threshold per group
       n candidates scores
**Output:** $i \in [n]$, index of chosen candidate
**for** $i \leftarrow 1$ **to** $n$ **do**
    **if** $\tau_{i'} > t_{c(i)}$ **then**
        **if** $i \succ \max \{i' \mid \tau_{i'} \leq \tau_i, c(i' |) = c(i)\}$ **then**
            **return** $i$
        **end**
    **end**
**end**

---

This threshold determines from which point the algorithm could pick a candidate. Once the thresholds are computed, they are used as input to Algorithm 1 along with the data. The algorithm will return its best candidate.

## 3.2 Prophet algorithm

In contrast to the secretary setting, the prophet algorithm knows the distribution that the scores are drawn from. Furthermore, all prophet experiments occur in a setting where each candidate is in a unique group. The group size is therefore one for every group. This constraint aims to create an algorithm that gives candidates in each arrival order the same probability of being picked. Each group represents a position in the queue and arrival order of the candidates in this problem is not random.

The original authors consider two settings in their paper: one where each candidate is drawn from a separate distribution, and one where the scores are i.i.d., pseudocode for both models are illustrated in Algorithms[1] 2 and 3.

---

**Algorithm 2** Fair General Prophet

**Input:** $F_1...F_n$, distributions
$q_1...q_n$, fair optimal pick probability
n candidates scores
**Output:** $i \in [n]$, index of chosen candidate
**for** $i \leftarrow 1$ **to** $n$ **do**
    **if** $v_i \geq F_i^{-1}\left(1 - \frac{q_i/2}{1-s/2}\right)$ **then**
        **return** $i$
    **end**
    $s \leftarrow s + q_i$
**end**

---

**Algorithm 3** Fair IID Prophet

**Input:** $F_1...F_n$, distributions
$q_1...q_n$, fair optimal pick probability
n candidates scores
**Output:** $i \in [n]$, index of chosen candidate
**for** $i \leftarrow 1$ **to** $n$ **do**
    **if** $v_i \geq F^{-1}\left(1 - \frac{2/3n}{1-2(i-1)/3n}\right)$ **then**
        **return** $i$
    **end**
**end**

---

## 3.3 Evaluation metrics

For evaluating the experiments, the authors set several metrics that reflect both the fairness and efficacy of their study. For the Secretary algorithm they report the number of candidates picked by each model, the number of times the chosen candidates correspond to the maximum value $max\ C_j$ within their group, and the probability of choosing the maximum $max\ C$ from the data. Meanwhile, the Prophet algorithms are compared based on the balance in selection rates across arrival order and the average value of the picked candidates.

## 4 Code implementation

Implementation of the experiments was done in Python, making use of the descriptions in the paper and the published code base [2]. The original authors conducted their experiments in C++. The code was factorized neatly into different files for the data, implementation of algorithms and experiments.

We were largely able to reproduce all of the code. However, three important elements of the code were lacking:

- the experiments on the prophet algorithms
- the production of plots and summary statistics of both experiments
- the data preprocessing for real datasets

Due to these issues, we were unable to review the exact settings of the experiments. This made it difficult to determine the reason behind different results in our reimplementation. Details are expanded on in the following section. As a consequence, we contacted the authors for further specifications of their approach.

---

[1]Since the original authors refer to Algorithm 2 as the Fair General Prophet and FairPA interchangeable, it makes sense for us to do the same.

[2]Original code: `https://github.com/google-research/google-research/tree/master/fairness_and_bias_in_online_selection`. Our full implementation is open-sourced and can be found on: `https://anonymous.4open.science/r/GbHqExFJUMM2jzct3LmeEpq`

93 Moreover, the naming of the baselines and proposed algorithms in the C++ code is inconsistent with the naming in the
94 paper. The original papers of the baselines were needed to figure out the used naming conventions.

95 Lastly, it is important to note that our implementation does not utilise GPUs or parallelisation because we have
96 sequential process. Therefore the results could not be speeded up using high performance clusters. As a result, one of
97 our experiments could not be run in a timely manner as it took over 40 hours for $1/5$ of the data.

## 5 Experimental setup

### 5.1 Secretary problem

**Data:** The paper uses four different datasets for the Secretary problem, out of which two are synthetic. For each dataset,
the algorithm runs 20,000 times.

For the first dataset we divide candidates into four groups with $10$, $100$, $1000$, and $10,000$ occurrences. The
probabilities $p$ are the same for all colors, namely ($p = .25$). In the second setting, this condition is changed and
group probabilities differ: $p = (.3, .25, .25, .2)$. Thirdly, the authors use a dataset of phone calls made by a Portuguese
banking institution [14]. For the purpose of this experiment, the score is the length of the phone call. The group
probabilities are set to be equal ($p = .2$). Lastly, the algorithm is tested on a dataset of influencers of the social network
'Pokec'[16]. The influencers' score is their number of followers and they are divided into five groups with equal
probability: ($p = .2$) for each group.

**Baselines:** The authors test their fair Secretary algorithm by contrasting it to two baselines. **Secretary algo-
rithm (SA)** computes the maximum score value assigned in the first $1/e$ part of the arrival sequence of the candidates.
After that, it compares the rest of the values with the aforementioned picked one and returns the maximum value across
the whole streamline. It does not consider a candidate's group. The **Single-color secretary algorithm (SCSA)** selects a
color proportional to the provided $p$ values and then considers only candidates of that color.

### 5.2 Prophet problem

**Data:** The prophet algorithms are tested on two synthetic datasets. In the original paper, each algorithm runs $50,000$
times. In the first experiment, $50$ samples are drawn from a uniform distribution $[0, 1]$. In the second experiment, $1000$
samples are drawn from a binomial distribution with $1000$ trials and probability of success of $1/2$.

**Baselines:** The authors compare their devised algorithms with four baseline algorithms: First, the **SC algo-
rithm** [15], which places a single threshold such that it finds a candidate 50% of the time. Second, the **EHKS
algorithm** [12] where each candidate is selected with probability $\frac{1}{n}$. Thirdly, the **CFHOV algorithm** [11], which uses
a succession of thresholds derived from the probabilities that candidates are accepted. Lastly, the **DP algorithm** [8] that
uses a differential equation to create thresholds. This last algorithm is excluded from their plots, as it is so unbalanced
that it distorts the readability of the plot.

During implementation we noticed that both the SC and EHKS algorithms were significantly quicker. This is due
to their property of using a constant calculated threshold for each run of the algorithm, instead of recalculating the
threshold after seeing each candidate.

## 6 Replication of results

### 6.1 Secretary problem

Figure 1 shows the results of the Secretary experiments from Python implementation. Our implemented algorithm is
equally fair and appears to pick the optimal candidate with roughly the same frequency in three out of four experiments.
The results from the original paper can be found in the Appendix (Figure 3).

The authors report the evaluation metrics as described in Section 3.3. Both the original results and our experiment's
metrics are illustrated in Table 1. Our scores are generally comparable to those in the original paper, with a margin of
just 3-4%.

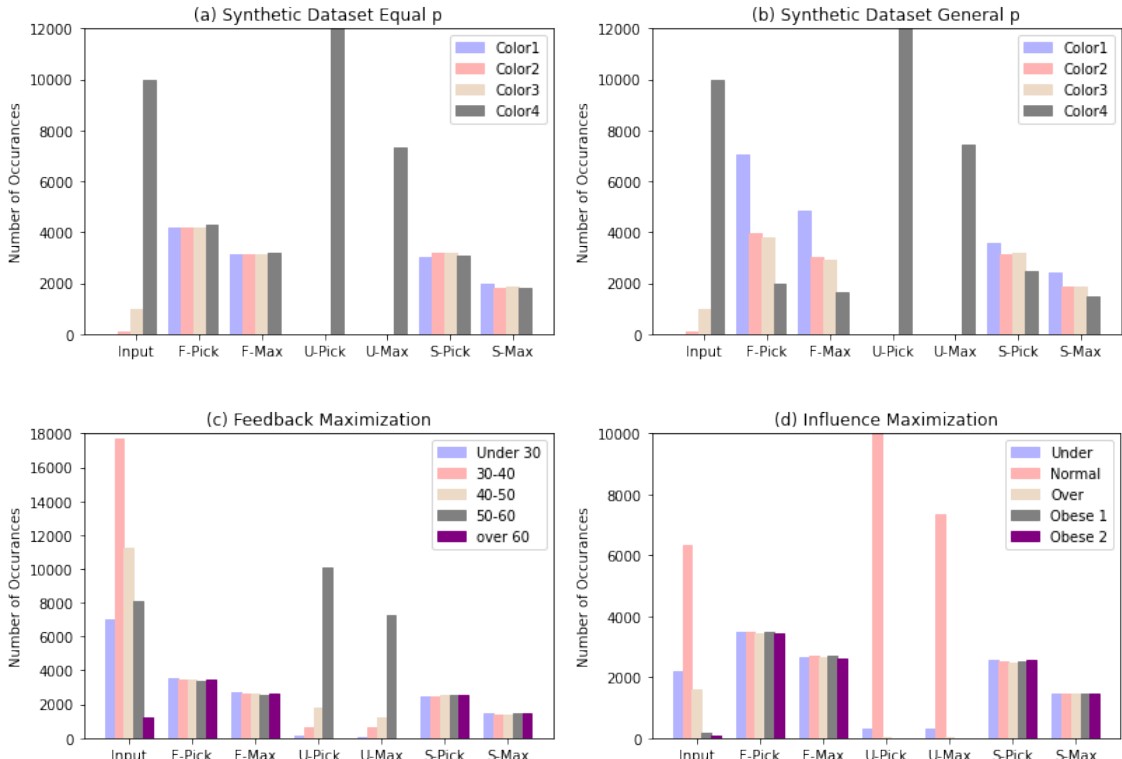

Figure 1: This plot contains our replication of the results for all four experiments, each one with a different dataset. Within each graph, there are 7 blocks of bars. The first one represents the group sizes of the colors we consider. The next two are called F-Pick and F-Max which illustrate the number of elements picked and the number of maximum among the elements picked by the Multi-Color Secretary Algorithm (MCSA). The same goes for U-Pick and U-Max which are representative of the plain Secretary Algorithm (SA). Lastly, S-Pick and S-Max illustrate the results of the Single-Color Secretary Algorithm (SCSA).

| | | Synthetic Data Equal p | Synthetic Data General p | Feedback maximization | Influence maximization |
|---|---|---|---|---|---|
| No. picked | Authors results | 1.305 | 1.309 | 1.347 | 1.373 |
| | Reproduced results | 1.354 | 1.350 | 1.372 | - |
| Max candidates | Authors results | 1.721 | 1.6302 | 1.760 | 1.756 |
| | Reproduced results | 1.684 | 1.636 | 1.837 | - |

Table 1: Evaluation metrics for the Secretary Problem. Each score represents how many more times the Multi-color Secretary Algorithm chose a candidate as compared to the most fair baseline, namely the Single-color Secretary algorithm.

The only algorithm in which our results differ is the SA, illustrated as U-Pick and U-Max. The SA algorithm created inconsistencies in two experiments. In the first case, the authors' experiments show that in setting (a) (synthetic dataset with equal $p$), SA almost exclusively returns candidates from group 4. However, in setting (b) (when p differs per group) it selects from multiple groups. This change is striking, as the SA algorithm should not take into account the probabilities of different groups.

We raised these observations in a chain of discussions with the authors. They confirmed our suspicions that the results of the SA algorithm are not intuitive, but did not know the reason for the discrepancies. They indicated that one possible reason would be the manner of sampling synthetic data. Given their explanation, we chose to further analyze their C++

implementation and figure out whether our results are incorrect or if there are other reasons for these differences. We found out that there are several inconsistencies in the C++ implementation as compared to the original paper:

- New synthetic data are generated for each of the 20,000 iterations of the experiment instead of using the same dataset for all iterations

- When testing whether the returned candidate $i$ is the maximum from its group $C_j$, the authors verify whether $i \in [max\ C_j - 10]$, where $maxC_j$ represents the maximum score of color $j$.

- Even though the original paper states that the probabilities $p$ are not taken into account by algorithm SA, the C++ implementation does take into account the probabilities when creating the synthetic data. It adds bias towards a certain group by assigning one candidate from that group the upper limit value of a $unsigned\ int$ 64 data type in C++ ( $2^{64} - 1$). This happens only in the second experiment.

Because of these implementation inconsistencies, parts of the experiments were not easily reproducible. We claim that they cause a difference in results for experiment (b). We test our claim by running the provided C++ implementation by altering the following inconsistencies: only generating data once, getting rid of the margin of 10, and not taking the probabilities $p$ into account when creating the data. The modified C++ implementation outputs results that are much closer to our findings. They are illustrated in the Appendix in Figure 5.

The SA algorithm also outputs different results for dataset (d) on Influence Maximization. The paper illustrates that U-pick selects candidates from two groups, namely *Under* and *Normal*. However, our algorithm picks candidates only from the *Normal* group. To reproduce the original results, our first step was to re-analyse the C++ implementation in detail to see whether there are any other inconsistencies we should consider. This was not the case for this experiment. The second step was to verify whether the inconsistencies come from the data. The authors included neither the data nor the preprocessing steps, so we used our own preprocessed data to run the C++ implementation. The results can be found in the Appendix in Figure 6[3]. Running our preprocessed data with our reproduced code and with the original c++ code yields exactly the same results. Therefore, the inconsistencies originate from the data itself, more precisely from how the authors compute the BMI scores and divide them between groups. [4] We can thus conclude that the Influence Maximization experiment would be exactly reproduced if we were to have access to the originally prepossessed data.

## 6.2 Prophet problem

Figure 2 shows the experiment outcomes of our replicated FairPA and FairIID algorithms. Even though the general trends in the plots match, our overall number of picks is far lower for each of the algorithms. The original authors appear to pick a candidate every single time.[5] In some cases, the number of picks even exceeds the number of experiments run, which should not be possible.[6]

By contrast, our reproduced FairPA algorithm returns a None-result 50% of the time, and FairIID 30%. This makes sense, as the algorithm was designed to pick each candidate with probability of $q/2$.[7] We also ran the original code, after making minor adjustments to get it running and to deal with None-picks. Running on this code shows identical results to our own reproduced implementation.

The most probable explanation for both discrepancies in the reproduced results is that the original authors ran $100,000$ experiments, instead of $50,000$. Figures 2c and 2d show that when running our algorithms with this number of experiments, the results are strongly comparable to those of the original authors. We also contacted the authors about this discrepancy. They confirmed that their reported number of picks was too high. Similarly to us, they assumed to have run the experiments $100k$ times instead of $50k$. However, they were not able to confirm this at the time. Since

---

[3]The authors ran approximately 1 billion iterations over the Pokec dataset. Due to time constraints we had to restrict our experiment to 20,000 iterations. For readability we also downsampled the size of the input

[4]For our experiment, we used the formula $BMI = weight/height^2$ and defined the health groups as described by the WHO.

[5]For instance the number of picked candidates per position for their FairPA algorithm in the uniform distribution is 1000. This corresponds to 50 positions x 1000 picks = 50,000 total picks, the same as the number of iterations used.

[6]The number of picks of the original FairIID algorithm in the uniform distribution dataset hovers around 1200, which would mean 1200 picks x 50 positions = 60,000 total picks, far more than the 50,000 iterations. The same is true for the FairIID algorithm under the binomial setting. The line is somewhat obscured by other algorithms, but appears to be consistently higher based on the reported number of experiments.

[7]$q$ is the probability of an optimal offline algorithm choosing a certain candidate. This offline algorithm has a none-rate of zero.

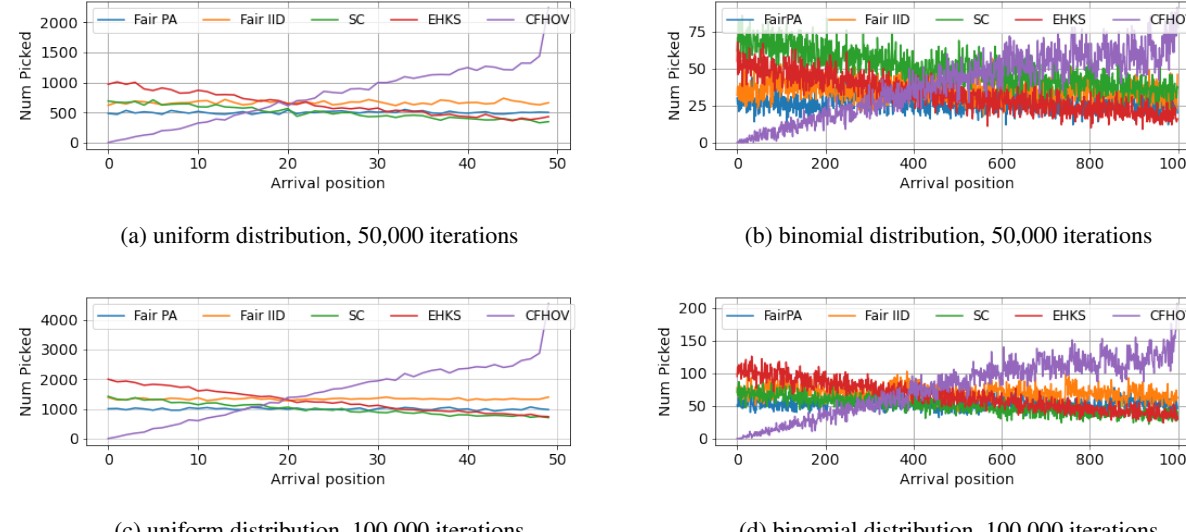

(a) uniform distribution, 50,000 iterations

(b) binomial distribution, 50,000 iterations

(c) uniform distribution, 100,000 iterations

(d) binomial distribution, 100,000 iterations

Figure 2: This plot contains our replication of the results for both prophet experiments, depicting the number of times the selected algorithms pick from each position in the candidate stream. Figure 2a & 2b refer to our experiment setup with 50,000 iterations, while Figure 2c & 2d refer to 100,000 iterations. The datasets for 2a and 2c are from the uniform distribution, while 2b and 2d come from the binomial distribution. Subfigures 2c and 2d are approximately identical to those in the original paper.

the implementation of the Prophet experiments themselves are not included in the code base, we were not able to definitively diagnose the reason for the discrepancy.

The original paper finds that the average values of chosen candidates for Algorithm 2 (FairPA), Algorithm 3 (FairIID), and the baselines SC, EHKS, CFHOV, and DP are 0.501, 0.661, 0.499, 0.631, 0.752, 0.751 respectively for the uniform distribution. For the binomial distribution, the values were 298.34, 389.24, 277.63, 363.97, 430.08, 513.34. Additionally, their FairPA and FairIID algorithms select candidates with an average value of "66.71% and 88.01% (for the uniform case), and 58.12% and 75.82% (for the binomial case), of the "optimal, but unfair, online algorithm" average.[8,9] They concluded that both proposed Algorithms 2 and 3 proved to be "ideally fair and seem to perform quite well in comparison to the best though unfair online algorithms." [9]

In our replication experiments, we found approximately the same results (deviation $< 1\%$) for these scores and percentages, under both the original 50k experiments and our proposed 100k experiments. However, we were only able to do this after assigning None-results with a value of zero in our implementation. Neither the ICML-version nor the full version of the paper mentions how their metrics deals with None-picks. Nevertheless, as taking None-picks as values of zero generated the same results, we assumed they used this approach.

## 7    Fair online decision making with higher pick-rates

As mentioned in section 6.2 the paper's proposed prophet algorithms only pick a candidate in only 50% of the cases. This is often not useful in practice. For this reason we extended on the original paper by adjusting the (mathematical) parameters used for the FairPA and FairIID algorithms. We contacted the authors to ask about their reasoning for using their parameters. They replied that their parameters achieved: "the best possible approximation ratio guarantee of a

---

[8]During communication with the authors it was brought to our attention that the results for the DP differ in the ICML version and the full version of the original paper, "due to a small issue in the calculation of the DP in the ICML version.". This results in the DP achieving an average score of 0.964 and 548.94 for the uniform and binomial distribution respectively. This then also changes the value of the optimal, but unfair algorithm to the following: " 51.97% and 68.57% (for the uniform case), and 54.35% and 70.91% (for the binomial case)." [10]. However, we focus on the ICML paper and thus focus on the presented results in this version. Partly due to the issue that no sufficient documentation could be found in order to solve this addressed issue in the DP algorithm.

[9]While the paper does not specify explicitly which unfair algorithm they mean in the paper, this seemed to refer to the DP algorithm.

1/2. However, they added: "It is possible that other algorithms also achieve the $1/2$ guarantee, or something close to it, while having other interesting properties, for instance being less wasteful."

Both algorithms 2 and 3 depend on calculating a top percentile that the candidate's value needs to be in. Each formula includes a constant of $1$. We change this constant to parameter $\epsilon$:

- The FairProphet algorithm depends essentially on $1 - \frac{q_i/2}{1-s/2}$. We change this fraction to $1 - \frac{q_i/2}{\epsilon-s/2}$

- The FairIID algorithm depends on $1 - \frac{2/3n}{1-2(i-1)/3n}$, which we change to $1 - \frac{2/3n}{\epsilon-2(i-1)/3n}$.

and perform a grid search to approximately find the optimal values.

We hypothesise that choosing a lower $\epsilon$ should decrease the top percentile a candidate needs to belong to in order to get selected. This should decrease the probability of finishing without picking any candidate, with the downside of achieving possibly a lower mean average value.

Our results for these grid-search experiments, including the used range, can be found in the Appendix section 8.3. Our updated version of both the FairPA and FairIID increases performance on all originally used metrics in the paper for both distributions, while seemingly still being fair. As $\epsilon$ decreases, the none-rate also goes down significantly. For the best found epsilon values, it even approaches zero. The algorithm remains approximately equally fair (see Appendix section 8.3)[10]. However, when $\epsilon$ becomes too low, the fairness starts to suffer. This is because the algorithm always chooses a candidate before getting to the end, meaning it never sees the last candidates in line. A change in $\epsilon$ also affects the average score. Excluding None results from the mean value, our optimal version performs slightly worse than the original authors' algorithm. This makes sense, as our algorithm is less picky and will also accept candidates with slightly lower scores. On the other hand, when including None-results as a $0$ value in the average, our algorithm outperforms the original authors'.

Looking at our proposed algorithm in terms of fairness it can be argued that our algorithm is fair. The proposed versions are perfectly fair in the sense that for every candidate, no matter wherever it arrives in the candidate stream, the chance of it being picked (if there is a pick anyway) is equal. However, we argue that if we are to define fairness in a different frame, namely both in the group and for each candidate, our algorithm is fair too. Since our versions pick more often a candidate, there is a higher change of a candidate being picked (which is fair in for distribution of goods as mentioned in the introduction), thus we raise the probability of selection for each candidate separately, which sums to a higher fairness for the whole group as a result, since not picking a candidate at all also is unfair for the whole group.

Lastly, an ablation study is irrelevant since removing an element is not needed or relevant for this algorithm, neither for our version nor the original presented one. Additionally we based the hyperparameter search range on having values above and below the original parameter.

# 8 Conclusion

To summarise this study: for both the Secretary and the Prophet problems we found that the results are largely reproducible. We did however find some inconsistencies in one of the baselines of the secretary problem, and on the scale of the prophet results. After further investigation, these discrepancies could be attributed to inconsistencies in the original authors' code. After this reproducibility study we conclude that the main claims made in the paper still hold.

The paper and the provided code base provided a good resource for reproducing the code. However, due to the absence of several parts of their code and the mentioned inconsistencies, the replication of the (exact) results took longer than expected. Fortunately, the authors showed to be very helpful and willing to answer our questions and concerns.

A drawback of the proposed prophet algorithms is that they only select a candidate in 50% (FairPA) and 30% (IID) of cases. Having such a None-result is often undesirable, so we introduced two adjusted prophet algorithms which have a pick rate of (close to) 100%. Our results suggest that these algorithms maintain similar levels of fairness.

As a point of discussion, we would like to note that knowing the group probabilities $p$ beforehand is, in some cases, quite counter intuitive. This fell outside of the scope of this reproducibility study, but it would be an interesting approach for further research to handle this critique.

---

[10]We would like to mention that we have not mathematically proven that our version is indeed 'fully' fair as the original authors did

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

# Appendix

## 8.1 Plots from original paper

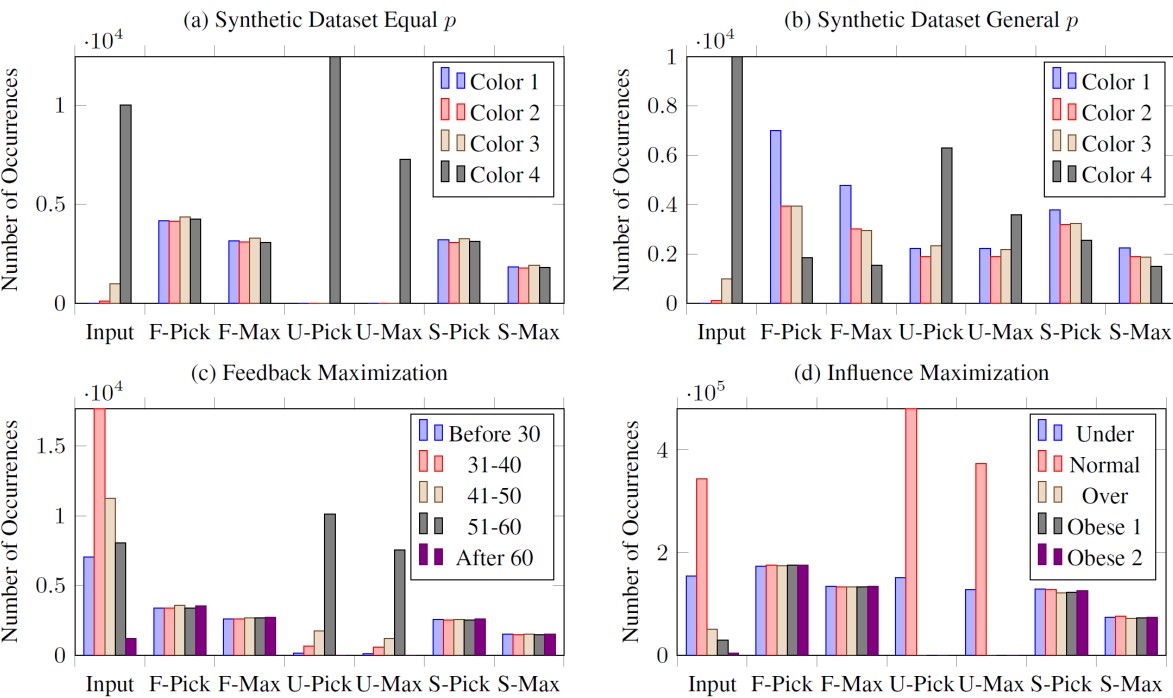

Figure 3: Results from the Secretary experiments in the original paper. The plot compares their fair secretary algorithm with the secretary algorithm (SA) and the single-color secretary algorithm (SCSA) on (a) synthetic dataset, equal p values, (b) synthetic dataset, general p values, (c) feedback maximization dataset, and (d) influence maximization dataset. Here Input is the number of elements from each color in the input, F-Pick and F-Max are the number of elements picked by our fair secretary algorithm and the number of them that are the maximum among the elements of that color. Similarly, U-Pick (S-Pick) and U-Max (S-Max) are the number of elements picked by SA and SCSA and the number of them that are the maximum among the elements of that color.

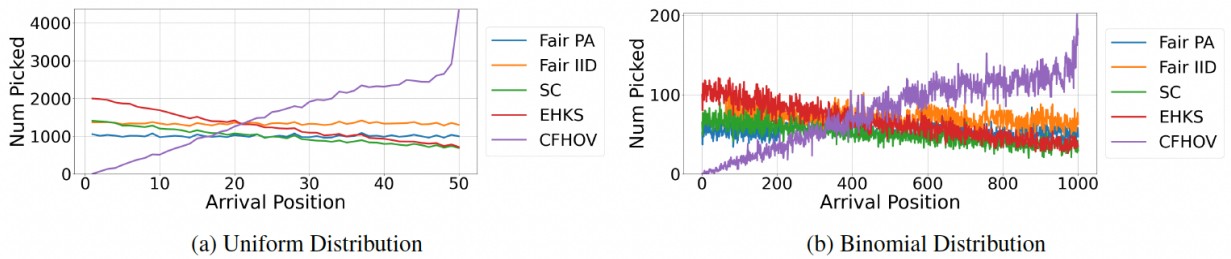

Figure 4: Results from the Prophet Experiments in the original paper: Results from the original paper. The plot represents the number of times that our algorithms (Fair PA, Fair IID) and the baselines (SC, EHKS, DP) pick from each position of the input prophet problem stream. In (a) the stream consists of 50 sample from the uniform distribution and in (b) the stream consist of 1000 sample from the binomial distribution.

 ## 8.2 Plots from original code

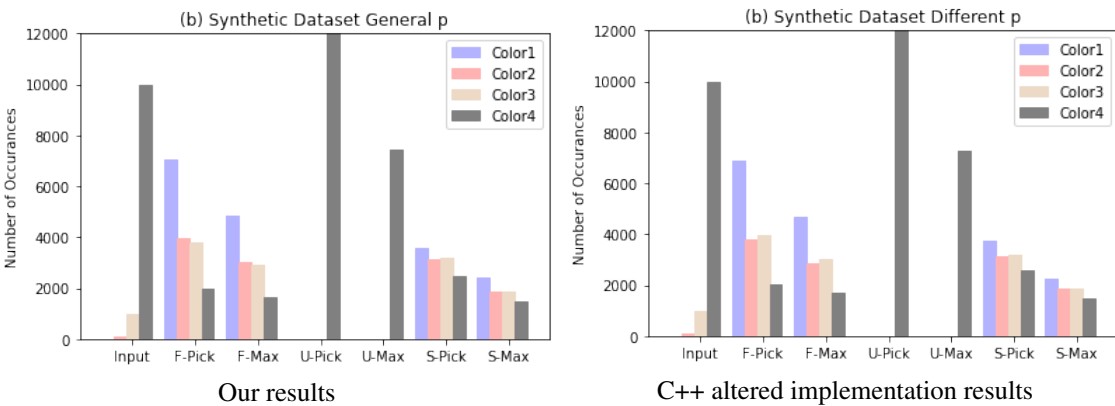

Our results

C++ altered implementation results

Figure 5: Results from running the original C++ code on the synthetic Dataset General p Experiment.

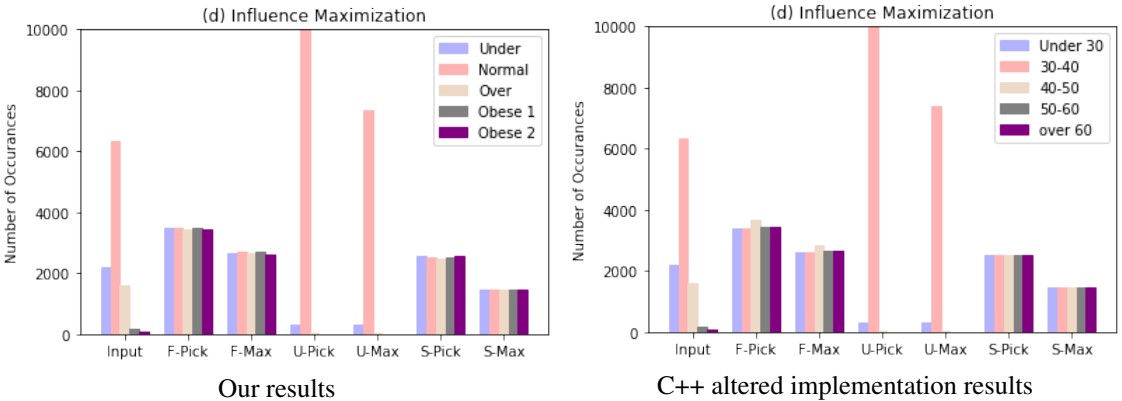

Our results

C++ altered implementation results

Figure 6: Results from running the original C++ code on the Influence Maximization Experiment.

 ## 8.3 Extension: parameter grid search

In this section, we present the results from our hyperparameter search on $\epsilon$ for both Fair Prophet algorithms. The original value of $\epsilon$ was 1.0 for both algorithms. As $\epsilon$ decreases, the None-rate goes down. Optimal values appear to be 0.5 for FairPA and 0.7 for the FairIID algorithm. If $\epsilon$ becomes lower than that, fairness suffers. The algorithm is then so unstrict, that it always makes a pick before seeing the last candidates in the queue.

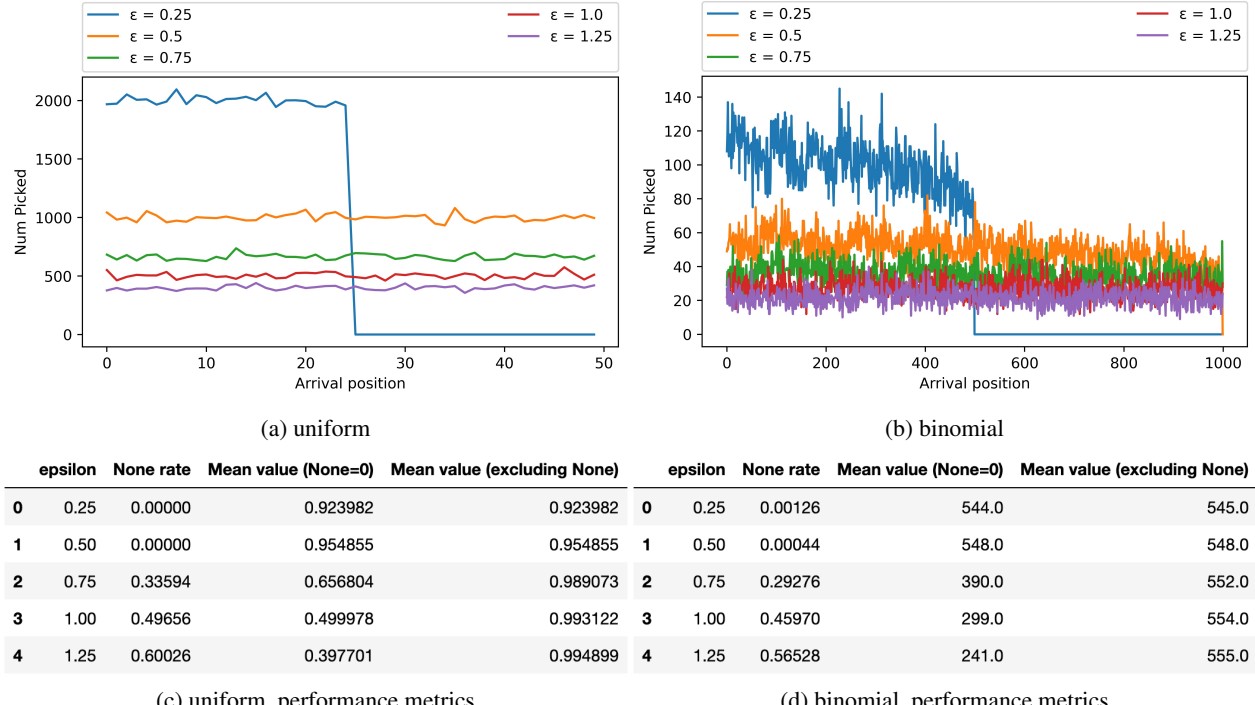

(a) uniform

(b) binomial

| | epsilon | None rate | Mean value (None=0) | Mean value (excluding None) | | epsilon | None rate | Mean value (None=0) | Mean value (excluding None) |
|---|---|---|---|---|---|---|---|---|---|
| **0** | 0.25 | 0.00000 | 0.923982 | 0.923982 | **0** | 0.25 | 0.00126 | 544.0 | 545.0 |
| **1** | 0.50 | 0.00000 | 0.954855 | 0.954855 | **1** | 0.50 | 0.00044 | 548.0 | 548.0 |
| **2** | 0.75 | 0.33594 | 0.656804 | 0.989073 | **2** | 0.75 | 0.29276 | 390.0 | 552.0 |
| **3** | 1.00 | 0.49656 | 0.499978 | 0.993122 | **3** | 1.00 | 0.45970 | 299.0 | 554.0 |
| **4** | 1.25 | 0.60026 | 0.397701 | 0.994899 | **4** | 1.25 | 0.56528 | 241.0 | 555.0 |

(c) uniform, performance metrics

(d) binomial, performance metrics

Figure 7: Results grid search of FairPA, as presented in section 7. Figures 7a & 7b show the number of picks per position for different values of $\epsilon$. The value $\epsilon = 1.0$ corresponds to the original paper's algorithm. Figures 7c & 7d show None rates and mean scores (including and excluding None results) for both settings.

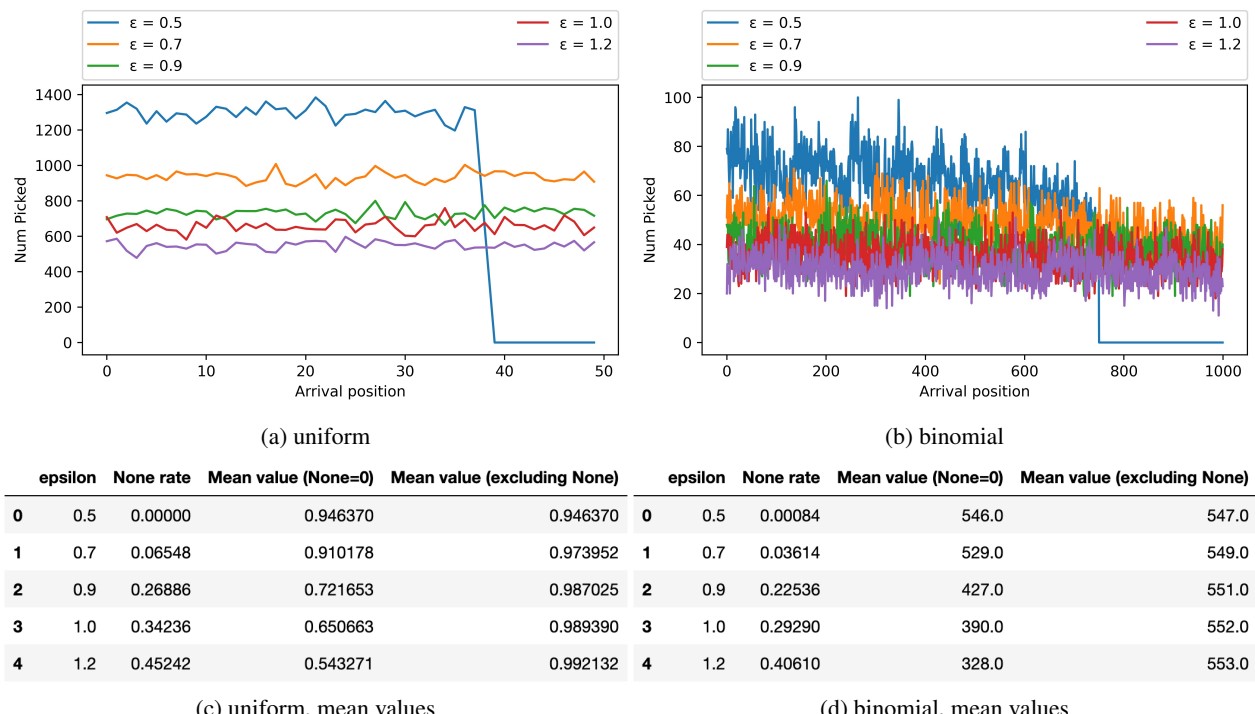

(a) uniform

(b) binomial

| | epsilon | None rate | Mean value (None=0) | Mean value (excluding None) | | epsilon | None rate | Mean value (None=0) | Mean value (excluding None) |
|---|---|---|---|---|---|---|---|---|---|
| **0** | 0.5 | 0.00000 | 0.946370 | 0.946370 | **0** | 0.5 | 0.00084 | 546.0 | 547.0 |
| **1** | 0.7 | 0.06548 | 0.910178 | 0.973952 | **1** | 0.7 | 0.03614 | 529.0 | 549.0 |
| **2** | 0.9 | 0.26886 | 0.721653 | 0.987025 | **2** | 0.9 | 0.22536 | 427.0 | 551.0 |
| **3** | 1.0 | 0.34236 | 0.650663 | 0.989390 | **3** | 1.0 | 0.29290 | 390.0 | 552.0 |
| **4** | 1.2 | 0.45242 | 0.543271 | 0.992132 | **4** | 1.2 | 0.40610 | 328.0 | 553.0 |

(c) uniform, mean values

(d) binomial, mean values

Figure 8: Results grid search of FairIID algorithm, as presented in section 7. Figures 8a & 8b show the number of picks per position for different values of $\epsilon$. The value $\epsilon = 1.0$ corresponds to the original paper's algorithm. Figures 8c & 8d show None rates and mean scores (including and excluding None results) for both settings.

