# OpenReview forum: "Replication Study of "Fairness and Bias in Online Selection""
_ML_Reproducibility_Challenge/2021/Fall — RC2021_

### Official Review · Reviewer_rgtT · 2022-02-23
**Review for Replication Study of "Fairness and Bias in Online Selection"**

**Rating:** 8
**Confidence:** 3

**Review:**

Scope of reproducibility

The report presents clearly the scope of reproducibility and adheres to it.

Code

The authors re-implement all algorithms from scratch. The code is submitted with a clear
guidance.

Communication with original authors

A fair communication is established with the original authors to inform the new findings with
an allegation regarding the inconsistencies between the original paper and the original code.

Hyperparameter Search

The authors use the hyperparameters as described in the original paper.

Discussion on results

The reproduction results are discussed clearly. The easy and difficult parts in the
reproducing are well presented.

Recommendations for reproducibility

The author comment on the inconsistencies between the original paper and the original
code. They also point out the drawback of the proposed prophet method in the original
paper.

Results beyond the paper

The authors introduce an adjusted algorithm to deal with the
disadvantage of the proposed prophet method in the original paper.

Overall organization and clarity

The report is well-written and has a good structure.

---

### Official Review · Reviewer_5tWR · 2022-03-01
**Review: Replication Study of "Fairness and Bias in Online Selection"**

**Rating:** 7
**Confidence:** 4

**Review:**

This replication study is clear, detailed, and convincing. The accompanying code is clear and well-documented. One suggestion for improvement would be to more clearly separate the discussion of the data generation process for the original paper from the discussion of the original paper's implementation of the algorithms. It sounds like (from the discussion on lines 155-169) that any inconsistencies between the reproduced result and the original paper was caused by the data generation process rather than the algorithmic component. Therefore, it is less helpful to refer to this as the C++ implementation, since the key component of the difference was the data generation aspect. Finally, it is not necessary to address the irrelevant parts of the replication analysis (e.g., the lack of an ablation study) - instead, a more thorough discussion of next steps would be helpful.

Minor points:
- There are some typographical errors that should be corrected, e.g., in Footnote 1 on page 3, "interchangeable" should be "interchangeably"; "speeded up" should be "sped up" on page 4,  "c++" should be "C++" on Line 166 of page 6, no hyphen in "ICML-version" on page 7, etc.
- In Table 1, "Authors" should be replaced with "Original" (or "Authors'" but "Original" is clearer).

---

### Meta-Review · Area_Chair_vfX9 · 2022-04-08

**Recommendation:** Accept
**Confidence:** 4

**Metareview:**

A great reproducibility study with a well-documented codebase. It would be great to see address some of the writing and presentation issues that reviewers pointed out.

---

### Decision · Program_Chairs · 2022-04-09

**Decision:**

Accept

**Comment:**

Following the recommendation of reviewers and meta-reviewer, the paper is accepted for ML Reproducibility Challenge 2021, and will be published in the upcoming special edition of ReScience Journal.